# Feature Space Translation Framework for Cross-Device Alignment and Derivation of Clinically Relevant Retinal Biomarkers from Foundation Model Embeddings

**Sparsh Rastogi** [† 1 2]   **Akshat Bakshi** [1]   **Jidugu Sriharisesh** [1]   **Imran Razzak** [3]   **Chinmaya Panigrahy** [1]   **Jatin Bedi** [1]   **Sahil Thakur** [4 5]

## Abstract

Oculomics enables non-invasive assessment of systemic diseases through retinal imaging, offering a portable and cost-effective alternative to traditional diagnostic methods. However, its clinical deployment is hindered by device-induced variability, which limits generalization across heterogeneous imaging systems. Although retinal foundation models have recently achieved strong performance across diverse downstream tasks, their robustness across heterogeneous imaging devices remains limited, and their latent representations do not directly correspond to clinically meaningful retinal biomarkers. To address these challenges, we propose a unified feature-space translation framework based on tabular learning that learns feature-space alignments between heterogeneous imaging devices and retinal foundation-model representations, enabling cross-device retinal biomarker translation and clinically meaningful biomarker estimation from foundation-model embeddings in both same-device and cross-device settings. Experiments on the AI-READI dataset demonstrate accurate cross-device retinal biomarker alignment, recovery of clinically meaningful biomarkers from RETFound embeddings, and robust cross-device biomarker estimation, achieving up to 78.50%, 87.50%, and 76.00% $R^2$, respectively. These findings establish feature-space translation as an effective strategy for mitigating device-induced variability while improving the interpretability and clinical utility of retinal foundation models without requiring architectural modifications.

## 1. Introduction

Oculomics is a field that leverages retinal imaging to enable the non-invasive assessment of systemic disease burden by capturing microvascular changes through structural biomarkers (Irodi et al., 2026; Zhu et al., 2025; Sung et al., 2026). The retina is the only external organ in the human body where microvasculature can be directly visualized non-invasively (Inamullah et al., 2025; Honavar, 2022), providing a unique view of blood vessels that feed blood to the heart, kidneys, brain and other vital organs. Changes in retinal biomarkers such as vessel tortuosity, fractal dimension, vessel density, and the arteriovenous ratio (AVR) have demonstrated strong diagnostic relevance for chronic conditions (Estrada et al., 2015; Huang & Dashtbozorg, 2018; Srinidhi et al., 2019; Wang et al., 2026), including chronic kidney disease (Sasongko et al., 2012; Tan et al., 2024a), type 2 diabetes, cardiovascular disease (Wong et al., 2002; Cheung et al., 2007; Rastogi et al., 2025), and neurological disorders (Liew et al., 2021; Blose & Silverstein, 2026; Chan et al., 2026). These biomarkers provide a unique window into systemic physiology, as retinal vasculature shares anatomical and functional similarities with other microvascular beds, making it particularly suitable for early disease detection and risk stratification (Xu & Trucco, 2026; Rastogi et al., 2025; Majithia et al., 2023; Lim et al., 2020; Tan et al., 2024b; Gupta et al., 2020; Cheung et al., 2020).

Recent advances in deep learning, along with standardized extraction pipelines such as AutoMorph (Zhou et al., 2022), VAMPIRE, IVAN, SIVA (McGrory et al., 2018) have enabled large-scale, automated, and reproducible quantification of these retinal features from fundus images (Fraz et al., 2015; Perez-Rovira et al., 2011). In parallel, retinal foundation models (FMs) pretrained on large-scale datasets (Gharbi et al., 2026), such as RETFound (Zhou et al., 2023), RetFound Green (Engelmann & Bernabeu, 2025), VisionFM

___

†Work conducted independently of Meesho. [1]Thapar Institute of Engineering and Technology, Patiala, India [2]Meesho, Bengaluru, India [3]Mohamed bin Zayed University of Artificial Intelligence (MBZUAI), Abu Dhabi, UAE [4]Mediwhale, Seoul, South Korea [5]Singapore Eye Research Institute, Singapore. Correspondence to: Sparsh Rastogi <srastogi_be22@thapar.edu>, Sahil Thakur <sahil.thakur@mediwhale.com>.

*Proceedings of the $43^{rd}$ International Conference on Machine Learning*, Seoul, South Korea. PMLR 306, 2026. Copyright 2026 by the author(s).

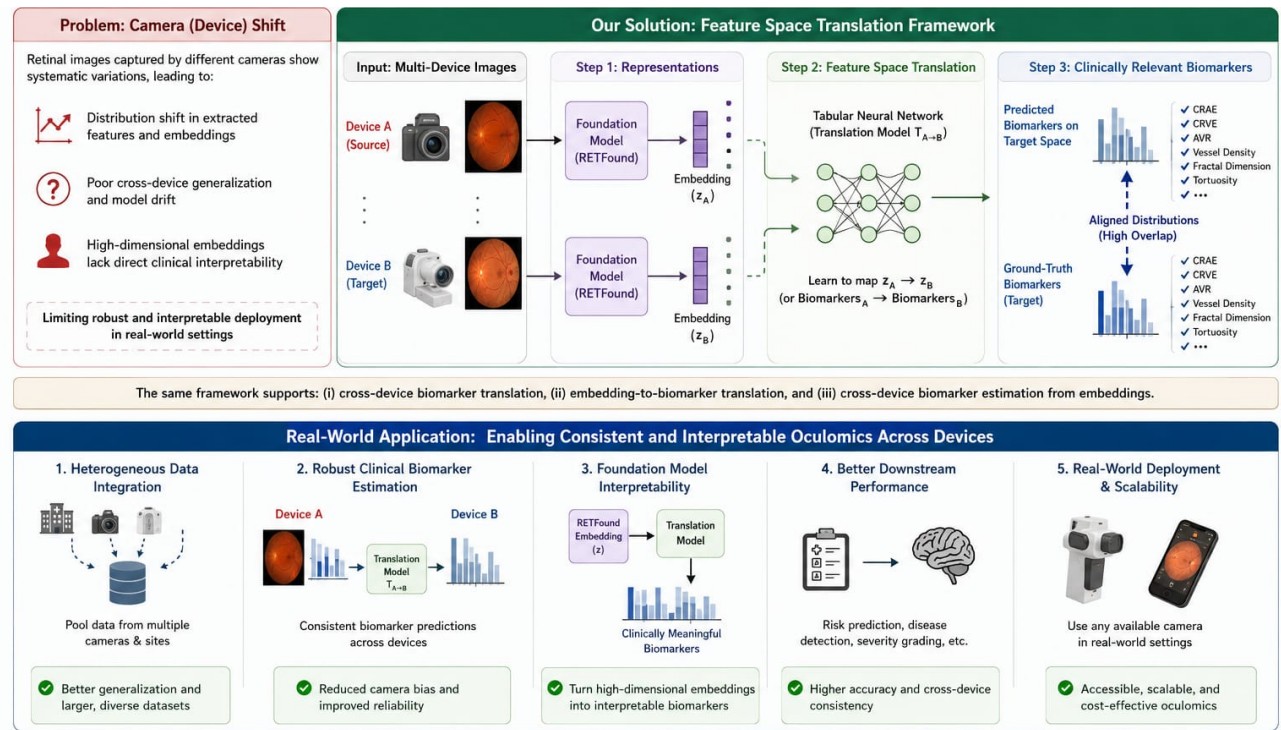

*Figure 1.* Overview of the proposed feature space translation framework. The training pipeline consists of three tasks: (i) cross-device biomarker translation, (ii) embedding-to-biomarker translation, and (iii) cross-device biomarker estimation from embeddings. The framework operates on retinal fundus images from multiple devices and is evaluated using both prediction accuracy and distributional alignment metrics.

(Qiu et al., 2023; 2024), EyeCLIP (Shi et al., 2025) have been proposed, leveraging extensive pretraining to shift the paradigm from handcrafted feature extraction to representation learning. These models learn rich, hierarchical representations that capture complex retinal patterns and achieve strong performance across diverse downstream tasks.

Despite this progress, the deployment of these approaches in real-world clinical settings remains limited due to two major constraints. First, out-of-distribution (OOD) shifts arising from camera and device heterogeneity introduce systematic variability in image characteristics and extracted features, thereby reducing robustness and leading to model drift across different devices, populations and datasets (Futoma et al., 2020; Cunnac et al., 2026). Although several studies have demonstrated robust generalization across population-level distribution shifts (Thakur et al., 2025; Tseng et al., 2023; Joo et al., 2023; Rim et al., 2021; Ting et al., 2017; Sabanayagam et al., 2020; Christopher et al., 2018), relatively limited work has focused on addressing distribution shifts introduced by differences in imaging devices and acquisition protocols (Rastogi & Thakur, 2025; Chen et al., 2024; Guo et al., 2024), which remain a major barrier to the development of generalizable and clinically deployable oculomic models (Cunnac et al., 2026; Kubin et al., 2024; Vilela et al., 2024; Abay et al., 2025; Ahn & Kim, 2024; Panwar

et al., 2015). Second, despite their strong representational capacity, the high-dimensional latent embeddings produced by these models lack direct correspondence with clinically interpretable retinal biomarkers, limiting their transparency, clinical validation, and broader adoption in practice (An et al., 2025; Mensah et al., 2026).

To overcome these limitations, we propose a feature space translation framework that leverages tabular neural networks to map retinal biomarkers from the feature space of one imaging device to another, thereby mitigating device-induced variability and camera drift. Rather than introducing a new architectural design, our approach focuses on learning cross-device correspondences in feature space to improve robustness across heterogeneous imaging systems. Furthermore, we extend this framework to operate on the latent embeddings of retinal foundation models, such as RETFound (Zhou et al., 2023) to translate high-dimensional representations into clinically meaningful retinal biomarkers. In addition, the same formulation enables cross-device biomarker estimation directly from embeddings, allowing representations derived from one device to be calibrated and interpreted in the context of another. By decoupling representation learning from device-specific biases, our approach provides a simple yet effective way to improve the consistency, interpretability, and cross-device applicability

of oculomics pipelines. The key contributions of this work could be summarized as follows:

- We present a unified feature-space translation framework that aligns retinal biomarkers across imaging devices, translates latent representations from retinal foundation models into clinically meaningful biomarkers, and enables cross-device biomarker estimation directly from foundation-model embeddings.

- We benchmark a diverse set of state-of-the-art tabular neural networks alongside classical machine learning methods for feature-space translation, providing a comprehensive evaluation of their effectiveness for cross-device retinal biomarker alignment.

- Our framework is model-agnostic and can be seamlessly integrated into existing oculomics pipelines without architectural modifications, improving cross-device consistency and biomarker interpretability with minimal computational overhead.

## 2. Methodology

**Problem Formulation.** Let $\mathcal{D} = \{d_1, \ldots, d_K\}$ denote a set of imaging devices. For each device $d \in \mathcal{D}$, let $X^{(d)} \in \mathbb{R}^p$ represent the structured retinal biomarker space (e.g., AutoMorph features), and $E^{(d)} \in \mathbb{R}^q$ denote the corresponding latent representation obtained from a retinal foundation model such as RETFound. We aim to learn a set of parametric mappings that enable (i) cross-device biomarker translation (Eq. 1), (ii) embedding-to-biomarker translation within a device (Eq. 2), and (iii) cross-device biomarker estimation directly from embeddings (Eq. 3).

$$f_{d_i \to d_j} : \mathbb{R}^p \to \mathbb{R}^p, \quad X^{(d_j)} \approx f_{d_i \to d_j}(X^{(d_i)}) \quad (1)$$

$$g_d : \mathbb{R}^q \to \mathbb{R}^p, \quad X^{(d)} \approx g_d(E^{(d)}) \quad (2)$$

$$h_{d_i \to d_j} : \mathbb{R}^q \to \mathbb{R}^p, \quad X^{(d_j)} \approx h_{d_i \to d_j}(E^{(d_i)}) \quad (3)$$

where $f_{d_i \to d_j}$, $g_d$, and $h_{d_i \to d_j}$ correspond to tasks (i), (ii), and (iii), respectively. All mappings are learned using tabular predictive models by minimizing supervised reconstruction objectives over paired samples. For a given mapping $\phi \in \{f_{d_i \to d_j}, g_d, h_{d_i \to d_j}\}$, we define a general objective (Eq. 4) over input–target pairs $(u, v) \sim \mathcal{P}$, where $\ell(\cdot, \cdot)$ denotes a task-specific loss function.

$$\mathcal{L}(\phi) = \mathbb{E}_{(u,v)\sim\mathcal{P}} \left[ \ell\big(v, \phi(u)\big) \right] \quad (4)$$

$$\mathcal{L}_{\text{NLL}}(\phi) = -\mathbb{E}_{(u,v)\sim\mathcal{P}} \left[ \log p_\phi\big(v \mid u\big) \right] \quad (5)$$

Under a probabilistic formulation, we model the conditional distribution $p_\phi(v \mid u)$ and minimize the negative log-likelihood (Eq. 5). Assuming a Gaussian likelihood with mean $\phi(u)$ and fixed variance, this reduces to a mean squared error objective. To account for distributional discrepancies induced by cross-device variability, we additionally consider a Wasserstein-distance-based objective (Eq. 6) that promotes alignment between the predicted and target distributions. The overall optimization problem is defined in Eq. 7, where the mappings are learned over paired samples. This formulation provides a unified framework for addressing device-induced variability while improving the interpretability of foundation model representations.

$$\mathcal{L}_{\text{W}}(\phi) = \mathbb{E}_{(u,v)\sim\mathcal{P}} \left[ W\big(\delta_{\phi(u)}, \delta_v\big) \right] \quad (6)$$

$$\min_\phi \mathcal{L}(\phi), \quad \phi \in \{f_{d_i \to d_j}, g_d, h_{d_i \to d_j}\} \quad (7)$$

**Data Curation.** We use retinal fundus images from the AI-READI dataset (AI-READI Consortium, 2024b;a), comprising 1,067 participants imaged using multiple non-mydriatic fundus cameras, primarily the Topcon Triton and Topcon Maestro2. The paired multi-device acquisitions enable supervised learning of cross-device feature mappings while capturing realistic acquisition variability. For each image, we extract 12 clinically relevant retinal vascular biomarkers using AutoMorph, including measures of vessel caliber (e.g., CRAE, CRVE, and AVR), vascular complexity (e.g., fractal dimension and vessel density), and geometric properties such as vessel tortuosity. We additionally extract high-dimensional latent representations using the pretrained RETFound model, yielding complementary structured biomarker vectors and foundation-model embeddings for each sample.

**Framework Overview.** The overall pipeline is illustrated in Fig. 1. Our framework comprises three complementary feature-space translation tasks: (i) cross-device biomarker translation by learning mappings between AutoMorph-derived biomarkers extracted from paired Triton and Maestro2 images; (ii) embedding-to-biomarker translation by learning device-specific mappings from RETFound embeddings to clinically meaningful retinal biomarkers; and (iii) cross-device biomarker estimation by predicting target-device biomarkers directly from source-device RETFound embeddings. We investigate these tasks using eight tabular predictive models spanning both classical machine learning (Linear Regression, Ridge Regression, Gradient Boosting, and CatBoost) and modern tabular neural networks, including TabPFN (Hollmann et al., 2023), TabM (Gorishniy et al., 2025), RealMLP, and a lightweight MLP. Together, these models represent a broad spectrum of inductive biases, ranging from simple linear baselines to recent architectures that

*Table 1.* Cross-device biomarker translation performance from Maestro2 to Triton across eight tabular learning models. **Bold** indicates the best result, while underline denotes the second-best.

| | TabPFN | | Lightweight MLP | | CatBoost | | TabM | | RealMLP | | Ridge | | Gradient Boosting | | Linear Regression | |
|---|---|---|---|---|---|---|---|---|---|---|---|---|---|---|---|---|
| Biomarker | $R^2$ (%) | RMSE | $R^2$ (%) | RMSE | $R^2$ (%) | RMSE | $R^2$ (%) | RMSE | $R^2$ (%) | RMSE | $R^2$ (%) | RMSE | $R^2$ (%) | RMSE | $R^2$ (%) | RMSE |
| AVR (Knudtson) | **59.70** | **0.05** | 59.20 | 0.05 | 53.00 | 0.05 | 59.20 | 0.05 | 56.80 | 0.05 | 57.80 | 0.05 | 50.30 | 0.05 | 57.80 | 0.05 |
| Artery Distance Tortuosity | **29.60** | **3.07** | 25.60 | 3.16 | 26.60 | 3.13 | 27.80 | 3.11 | 24.60 | 3.18 | 25.60 | 3.16 | 22.10 | 3.23 | 25.30 | 3.16 |
| Artery Fractal Dimension | **68.80** | **0.03** | 66.90 | 0.03 | 63.30 | 0.03 | 68.30 | 0.03 | 68.20 | 0.03 | 67.50 | 0.03 | 64.50 | 0.03 | 67.50 | 0.03 |
| Average Width | **67.40** | **3.58** | 66.40 | 3.63 | 61.00 | 3.91 | 67.30 | 3.58 | 64.80 | 3.72 | 66.80 | 3.61 | 64.90 | 3.71 | 66.70 | 3.62 |
| CRAE (Knudtson) | 72.90 | 11.39 | 73.50 | 11.28 | 72.50 | 11.48 | 72.30 | 11.52 | **73.70** | **11.22** | 73.00 | 11.37 | 69.80 | 12.02 | 73.00 | 11.37 |
| CRVE (Knudtson) | 78.20 | 14.38 | 78.00 | 14.46 | 76.80 | 14.86 | 77.70 | 14.56 | 78.10 | 14.43 | **78.50** | **14.30** | 74.80 | 15.48 | 78.40 | 14.32 |
| Distance Tortuosity | **42.90** | **0.01** | 41.20 | 0.01 | 39.50 | 0.01 | 42.00 | 0.01 | 40.50 | 0.01 | 41.50 | 0.01 | 37.20 | 0.01 | 41.30 | 0.01 |
| Fractal Dimension | **70.10** | **0.02** | 69.40 | 0.02 | 67.50 | 0.02 | 69.90 | 0.02 | 69.20 | 0.02 | 69.70 | 0.02 | 66.10 | 0.02 | 69.60 | 0.02 |
| Tortuosity Density | **51.20** | **0.02** | 49.80 | 0.02 | 47.00 | 0.02 | 50.50 | 0.02 | 48.80 | 0.02 | 50.00 | 0.02 | 45.20 | 0.02 | 49.90 | 0.02 |
| Vein Distance Tortuosity | **31.80** | **0.01** | 30.20 | 0.02 | 28.10 | 0.02 | 31.00 | 0.01 | 29.50 | 0.02 | 30.50 | 0.01 | 26.00 | 0.02 | 30.30 | 0.02 |
| Vein Fractal Dimension | **64.50** | **0.02** | 63.20 | 0.02 | 60.10 | 0.02 | 64.00 | 0.02 | 62.80 | 0.02 | 63.60 | 0.02 | 59.00 | 0.02 | 63.50 | 0.02 |
| Vessel Density | **58.90** | **0.03** | 57.50 | 0.03 | 54.20 | 0.03 | 58.40 | 0.03 | 56.70 | 0.03 | 57.90 | 0.03 | 53.00 | 0.03 | 57.70 | 0.03 |

achieve state-of-the-art performance on tabular benchmarks such as TabArena ([Erickson et al., 2025](#)).

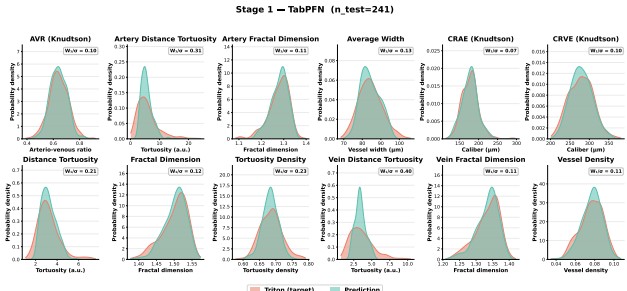

*Figure 2.* Distributional alignment for cross-device retinal biomarker translation from Maestro2 to Triton. Predicted biomarker distributions closely match the target distributions for key vascular biomarkers, demonstrating effective mitigation of device-induced variability.

**Evaluation.** We evaluate our framework across three complementary feature-space translation settings. First, we perform cross-device biomarker translation by learning mappings between AutoMorph-derived biomarkers extracted from paired Triton and Maestro2 images. Second, we perform embedding-to-biomarker translation by learning device-specific mappings from RETFound embeddings to the corresponding AutoMorph biomarkers, evaluating the extent to which retinal foundation-model representations encode clinically meaningful information. Third, we perform cross-device biomarker estimation by predicting target-device biomarkers directly from source-device RETFound embeddings, enabling device-invariant biomarker estimation without requiring target-device imaging. Models are trained on paired samples from participants imaged using multiple devices. The dataset is split at the patient level into training (70%), validation (15%), and test (15%) sets, ensuring subject-disjoint partitions. Models are selected using 5-fold cross-validation on the training set. Performance is evaluated using mean absolute error (MAE) and coefficient of determination ($R^2$), while cross-device distributional alignment is quantified using the Wasserstein distance, Kullback–Leibler (KL) divergence, and Jensen–Shannon

(JS) divergence, providing a comprehensive assessment of both predictive accuracy and feature-space alignment.

## 3. Results and Discussion

**Cross-device Biomarker Translation.** Table 1 summarizes the results for cross-device biomarker translation from Maestro2 to Triton. Overall, the proposed framework achieves strong predictive performance, with morphological biomarkers such as CRAE, CRVE, fractal dimension, average vessel width, and vessel density attaining $R^2$ values of up to 78.50%, while also exhibiting close agreement between the predicted and target distributions with low Wasserstein distances (Fig. 2). In contrast, tortuosity-related biomarkers achieve lower $R^2$ values and show larger distributional discrepancies, suggesting greater sensitivity to device-specific acquisition variability. TabPFN consistently achieves the best or near-best performance across most biomarkers, while TabM and the lightweight MLP also perform competitively, highlighting the advantage of modern tabular learning methods over classical machine learning models for capturing complex cross-device relationships. Based on these observations, we restrict subsequent experiments to TabPFN, lightweight MLP, and CatBoost. Together, these results demonstrate that the proposed framework preserves both prediction accuracy and the underlying biomarker distributions, effectively mitigating device-induced variability.

*Table 2.* Performance of embedding-to-biomarker translation on Maestro2, mapping RETFound embeddings to AutoMorph-derived retinal biomarkers. **Bold** indicates the best result, while underline denotes the second-best.

| | TabPFN | | Ridge | | ElasticNet | | PLS | |
|---|---|---|---|---|---|---|---|---|
| Biomarker | $R^2$ (%) | RMSE | $R^2$ (%) | RMSE | $R^2$ (%) | RMSE | $R^2$ (%) | RMSE |
| AVR (Knudtson) | 5.80 | 0.08 | 2.70 | 0.08 | 3.30 | 0.08 | **8.30** | **0.08** |
| Artery Distance Tortuosity | -3.60 | 3.91 | -34.20 | 4.45 | 0.00 | 3.84 | **1.90** | **3.80** |
| Artery Fractal Dimension | 74.90 | 0.02 | 75.10 | 0.02 | **78.90** | **0.02** | 75.40 | 0.02 |
| Average Width | 47.50 | 4.81 | **57.60** | **4.33** | 53.90 | 4.51 | 42.60 | 5.03 |
| CRAE (Knudtson) | 48.70 | 16.71 | **53.70** | **15.88** | 50.40 | 16.43 | 51.10 | 16.32 |
| CRVE (Knudtson) | 49.00 | 27.98 | **54.00** | **26.56** | 50.10 | 27.68 | 53.90 | 26.61 |
| Distance Tortuosity | **23.30** | **0.82** | 12.60 | 0.88 | 18.10 | 0.85 | 12.90 | 0.87 |
| Fractal Dimension | 85.70 | 0.02 | **87.50** | **0.01** | 83.50 | 0.02 | 83.00 | 0.02 |
| Tortuosity Density | 24.10 | 0.03 | -3.50 | 0.03 | 14.90 | 0.03 | **24.30** | **0.03** |
| Vein Distance Tortuosity | -4.60 | 1.44 | -15.90 | 1.52 | **3.90** | **1.38** | -3.10 | 1.43 |

*Table 3.* Performance of cross-device biomarker estimation from RETFound embeddings (Maestro2 → Triton). RETFound embeddings are translated to retinal biomarkers and subsequently aligned across imaging devices. **Bold** indicates the best result, while underline denotes the second-best.

| Biomarker | TabPFN $R^2$ (%) | RMSE | CatBoost $R^2$ (%) | RMSE | Lightweight MLP $R^2$ (%) | RMSE | Ridge $R^2$ (%) | RMSE |
|---|---|---|---|---|---|---|---|---|
| AVR (Knudtson) | 4.20 | 0.09 | 1.70 | 0.09 | **5.20** | **0.09** | 4.00 | 0.09 |
| Artery Distance Tortuosity | 6.50 | 3.84 | 2.30 | 3.92 | -0.60 | 3.98 | **8.50** | **3.79** |
| Artery Fractal Dimension | 57.30 | 0.03 | 57.00 | 0.03 | 56.70 | 0.03 | **57.70** | **0.03** |
| Average Width | 31.80 | 5.13 | **33.80** | **5.05** | 33.10 | 5.08 | 32.20 | 5.11 |
| CRAE (Knudtson) | 44.30 | 18.58 | 44.20 | 18.59 | **46.10** | **18.26** | 45.20 | 18.42 |
| CRVE (Knudtson) | 44.50 | 27.07 | 44.20 | 27.13 | 43.70 | 27.25 | **54.00** | 26.56 |
| Distance Tortuosity | 21.60 | 0.79 | 19.10 | 0.80 | 11.60 | 0.83 | **19.60** | **0.80** |
| Fractal Dimension | 71.50 | 0.02 | 70.30 | 0.02 | **71.60** | 0.02 | 70.70 | 0.02 |
| Tortuosity Density | 19.40 | 0.03 | 17.60 | 0.03 | 14.70 | 0.03 | **19.50** | **0.03** |
| Vein Distance Tortuosity | -8.10 | 1.45 | -11.00 | 1.47 | -7.10 | 1.44 | **-6.00** | 1.44 |
| Vein Fractal Dimension | 71.90 | 0.02 | 69.90 | 0.02 | **73.10** | **0.02** | 71.90 | 0.02 |
| Vessel Density | **76.00** | 0.01 | 74.70 | 0.01 | 75.20 | **0.01** | 75.00 | 0.01 |

**Embedding-to-Biomarker Translation.** Table 2 summarizes the results for embedding-to-biomarker translation on Maestro2. Overall, RETFound embeddings capture substantial clinically meaningful retinal information, with biomarkers such as fractal dimension, artery fractal dimension, and vessel caliber (CRAE and CRVE) achieving $R^2$ values exceeding 75.00%. In contrast, tortuosity-related biomarkers are recovered less accurately, suggesting that these features are less explicitly encoded in the latent representation. This trend is consistent with the distributional analysis in Fig. 3, where biomarkers with higher $R^2$ values exhibit closer agreement between the predicted and target distributions and lower Wasserstein distances, while tortuosity measures show comparatively larger distributional discrepancies. Overall, these findings demonstrate that RETFound embeddings preserve clinically relevant vascular morphology, although the fidelity of the encoded information varies across biomarker types.

**Cross-device Biomarker Estimation from Embeddings.** Table 3 summarizes the results for compositional cross-device biomarker estimation from RETFound embeddings. As expected, performance decreases relative to the individual translation stages due to error propagation across the embedding-to-biomarker and cross-device translation mappings. Nevertheless, structural biomarkers, including fractal dimension, vessel density, and vessel caliber (CRAE and CRVE), retain strong predictive performance, indicating that clinically relevant vascular information is preserved throughout the compositional pipeline. This trend is further reflected in the distributional analysis (Fig. 4), where these biomarkers exhibit close agreement between the predicted and target distributions with comparatively low Wasserstein distances, whereas tortuosity-related biomarkers show larger distributional discrepancies. Together, these findings demonstrate that the proposed framework enables robust cross-device biomarker estimation directly from retinal foundation-model embeddings, supporting its potential for clinically interpretable and device-agnostic oculomics.

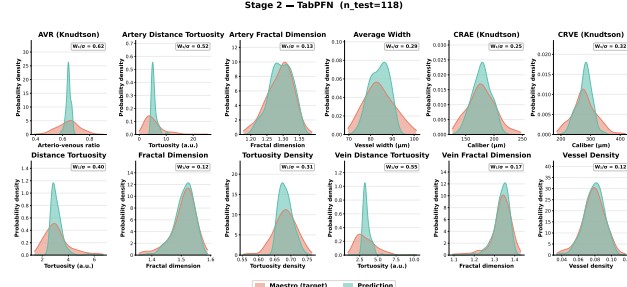

*Figure 3.* Distributional alignment for embedding-to-biomarker translation. Strong agreement between predicted and target biomarker distributions indicates that RETFound embeddings encode clinically meaningful retinal biomarkers.

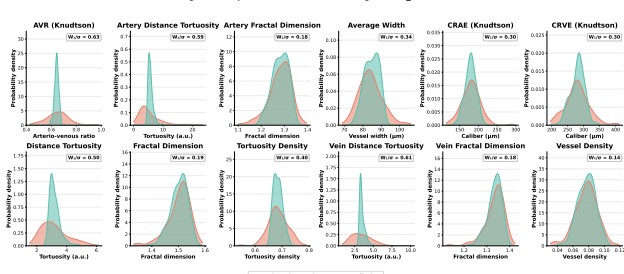

*Figure 4.* Distributional alignment for cross-device biomarker estimation from RETFound embeddings. Predicted distributions remain well aligned with target Triton biomarkers for major retinal biomarkers, demonstrating preservation of clinically meaningful information across the compositional translation pipeline.

## 4. Conclusion

In this work, we introduced a unified feature-space translation framework for addressing device-induced variability in oculomics. The proposed framework unifies cross-device retinal biomarker translation with the translation of retinal foundation-model embeddings into clinically meaningful biomarkers, enabling robust biomarker estimation across both same-device and cross-device settings. Experiments on the AI-READI dataset demonstrate that the learned mappings preserve both predictive accuracy and biomarker distributions, establishing feature-space translation as an effective strategy for improving cross-device robustness while enhancing the clinical interpretability of retinal foundation models. Future work will extend the framework to bidirectional and multi-device translation, evaluate it on the expanded AI-READI release and additional retinal datasets, and improve robustness to lower-quality imaging to further support large-scale, device-agnostic clinical deployment.

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
