# OpenReview forum: "Feature Space Translation Framework for Cross-Device Alignment and Derivation of Clinically Relevant Retinal Biomarkers from Foundation Model Embeddings"
_ICML.cc/2026/Workshop/FMSD — FMSD @ ICML 2026 Poster_

### Official Review · Reviewer_tp72 · 2026-05-19
**Feature Space Translation Framework for Cross-Device Alignment and Derivation of Clinically Relevant Retinal Biomarkers from Foundation Model Embeddings**

**Rating:** 6
**Confidence:** 4

**Review:**

Summary:

This paper proposes a feature-space translation framework to mitigate cross-device variability in retinal oculomics and to improve the interpretability of retinal foundation model (FM) embeddings. Using paired images from two devices in AI-READI (Topcon Maestro2 and Triton), the authors train tabular models to: (i) translate AutoMorph-derived biomarkers across devices, (ii) map RETFound embeddings to clinically meaningful biomarkers, and (iii) estimate target-device biomarkers from source-device embeddings via a compositional mapping. Results show moderate-to-strong R² for several vascular biomarkers (notably fractal dimension, vessel density, and vessel calibers) and improved distributional alignment, while tortuosity-related features remain challenging.

Pros:

1. Leverages a simple, model-agnostic tabular learning formulation to harmonize device-specific biomarker spaces without modifying image encoders or segmentation pipelines.

2. Proposes a unified view linking three practical tasks (cross-device biomarker translation, embedding-to-biomarker mapping, and cross-device estimation from embeddings), which could slot easily into existing oculomics workflows.

3. Emphasizes interpretability by regressing high-dimensional FM embeddings to clinically recognized biomarkers rather than only to task labels.


Cons:

1. The method relies on paired cross-device samples, limiting applicability when such pairs are scarce; no strategy for unpaired or partially paired settings is presented.

2. The Wasserstein-based objective is introduced but it is unclear whether it is actually used in training (beyond evaluation), and no ablation isolates its contribution.

3. The compositional mapping (embedding→biomarkers→target device) is evaluated, but the directly defined $h_{d_i→d_j}(E^{(d_i)})$ mapping (Eq. 3) is not reported; this misses a potentially stronger, single-step baseline.